# Research on Niche Evaluation of Photovoltaic Agriculture in China

**DOI:** 10.3390/ijerph192214702

**Published:** 2022-11-09

**Authors:** Jian Chen, Lingjun Wang, Yuanyuan Li

**Affiliations:** 1College of Economics and Management, Nanjing Forestry University, Nanjing 210037, China; 2School of Economics and Management, Nanjing Institute of Technology, Nanjing 211167, China; 3School of Food Science, Nanjing Xiaozhuang University, Nanjing 211171, China

**Keywords:** photovoltaic agriculture, niche evaluation, rough set, DANP, cloud model

## Abstract

To evaluate the ecological niche of China’s photovoltaic agriculture, this paper firstly analyzed the composition of photovoltaic agriculture and constructed the ecosystem of photovoltaic agriculture. Then, we defined the concept of the ecological niche of photovoltaic agriculture, and based on this the preliminary niche evaluation index system was constructed. Further, redundant indicators in the preliminary index system were deleted based on the rough set theory, and the final niche evaluation index system was constructed. Finally, the ecological niche of photovoltaic agriculture was evaluated using the DANP method and cloud model. We found that the niche level of China’s photovoltaic agriculture is between low and medium levels. Specifically, the level of resource niche is the highest, between medium and high levels; following is policy niche, near medium level; then is environmental niche, which is at a slightly lower medium level; the last three in turn are technology niche, social niche and economic niche. The technology should fully realize the synergistic effect of photovoltaic power generation and agricultural production, and the policy should play better environmental, social and economic functions on this basis to achieve a higher niche level of China’s photovoltaic agriculture.

## 1. Introduction

The combination of photovoltaic and agriculture appeared earlier in the field of agricultural irrigation. Katzman and Matlin took Nebraska and Texas in the United States as examples to analyze the cost-effectiveness and commercial feasibility of a photovoltaic irrigation system. They predicted that photovoltaic applications in agriculture would be more available after 2000 [1]. In the 21st Century, although the literature still mentioned the application obstacle of photovoltaic in agriculture at the early stage [2,3]. But now, the application scale of photovoltaic in agriculture is gradually expanding, and showing a trend of diversification [4], such as agricultural irrigation [5], refrigeration and drying of agricultural products [6], agricultural electric vehicles [7], farm lighting [8], agricultural greenhouse system [9], etc. Among them, due to the importance of water in agricultural production, the photovoltaic application in agricultural irrigation is concerned [10,11,12,13,14,15,16]. In addition, the application of photovoltaic in facility agriculture such as agricultural greenhouse systems has become a major recent research hotspot [17,18,19].

Goetzberger and Zastrow proposed the idea that solar conversion and crop cultivation can coexist, which specifically refers to modifying solar power plants to enable land to be used simultaneously for crop production, and they believed that further technological improvements could improve the applicability of solar power plants in crop production [20]. About 30 years later, Dupraz et al. innovatively put forward the concept of “agrivoltaic” [21]. They found that photovoltaic panels set up block the sunlight for crop production, thus reducing the crop yield. But at the same time, photovoltaic panels shade reduces the transpiration, and may improve the use and efficiency of the water. The key is to find a balance between photovoltaic power generation and crop production. Marrou et al. conducted follow-up studies along the above ideas, and found that the erection of photovoltaic panels reduced the steam dispersion of water in crops, which could increase crop yield, but required choosing suitable crop varieties, such as lettuce and other shade-loving crops [22]. At the same time, the agrivoltaic system can also be optimized through the special arrangement of photovoltaic panels to find the best collaboration between photovoltaic power generation and crop production. Scholars made optimistic predictions about the application prospect of agrivoltaic systems. Harinarayana and Vasavi believed that the technology can be successfully implemented in India [23]. In Germany, people will be happy to accept this dual use of land because it will not reduce the area of crops [24]. In the Phoenix metropolitan statistical area of the United States, the application of agrivoltaic systems can not only meet the social demand for clean electricity, but also protect the surrounding agricultural production land, so it is recommended to introduce agrivoltaic technology in this area [25]. Meanwhile, with the start of many agrivoltaic projects worldwide [26], agrivoltaics reflect a good economy [27,28,29], social [30,31], and environmental benefits [25,32,33]. 

Photovoltaic agriculture includes the above two aspects. This concept is unique to China [34], and we defined it exactly in Section 2. In recent years, China has gradually strengthened its research in the field of photovoltaic agriculture, and has carried out large-scale and diversified project practices [26,29]. However, due to the lack of official statistical information on the development of photovoltaic agriculture, the academic community is lack of understanding of the specific implementation of photovoltaic agriculture, resulting in the inability to assess the environmental resource consumption and the environmental, economic, and social impact, and thus unable to formulate appropriate management plans to promote the sustainable development of photovoltaic agriculture [35]. 

Niche is a key concept in ecology, and it refers to the position of certain species in a multidimensional space that is formed with environmental resources or conditions as gradients [36]. The ecological niche of photovoltaic agriculture refers to the position in time and space, and the relationship and function within the system or with other industries. It is the platform for communication between photovoltaic agriculture and the environment [34]. Niche theory emphasizes the selection and influence of the environment on organizations. The research on the ecological niche of photovoltaic agriculture has a great reference for the survival and development of photovoltaic agriculture. 

This paper tried to collect and analyze the relevant data on photovoltaic agriculture, evaluate the ecological niche of photovoltaic agriculture, generally grasp the resource occupation of photovoltaic agriculture, and evaluate the various impacts of photovoltaic agriculture in the environmental, economic and social aspects. We regarded photovoltaic agriculture as a special industrial organization, and studied the development status of photovoltaic agriculture from the perspective of social organisms. The research results are conducive to the sustainable development of photovoltaic agriculture, and can also play a policy enlightenment role on how to use photovoltaic agriculture for resource allocation in China. 

The main contributions of this paper are as follows: (1) The concept of photovoltaic agriculture was defined, and based on this the preliminary niche evaluation index system was constructed. (2) A final niche evaluation index system of photovoltaic agriculture was constructed based on the rough set theory. (3) The weight of each niche evaluation indicator was determined by the DANP method, and the niche level of photovoltaic agriculture was evaluated using a cloud model. 

The rest of the paper is organized as follows. The following section preliminarily constructs the index system of niche evaluation of photovoltaic agriculture. In Section 3, methods are presented. Section 4 summarizes the results. In Section 5, the results are discussed and the conclusions are presented. 

## 2. Preliminary Construction of Niche Evaluation Index System

### 2.1. Analytical Framework

To evaluate the ecological niche of photovoltaic agriculture, this paper first analyzed the composition of photovoltaic agriculture and constructed the ecosystem of photovoltaic agriculture. Then, the concept of the ecological niche of photovoltaic agriculture was defined, and based on this the preliminary niche evaluation index system was constructed. Further, redundant indicators in the preliminary index system were deleted based on the rough set theory, and the final niche evaluation index system was constructed. Finally, the ecological niche of photovoltaic agriculture was evaluated by DANP method and cloud model. The analytical framework is shown in Figure 1. 

### 2.2. The Ecological Niche of Photovoltaic Agriculture

#### 2.2.1. Photovoltaic Agriculture

Photovoltaic agriculture includes applications of photovoltaic in agriculture and agrivoltaics. The former is to supply all the photovoltaic power generation for agricultural production, while the latter is to supply agricultural production first (generally) and then connect the remaining power to the grid. So, conceptually, the latter includes the former, and applications of photovoltaic in agriculture can be embedded in agrivoltaics (See Figure 2). 

It can be seen from Figure 2 that the application of photovoltaic in agriculture reflects the combination of photovoltaic power generation technology and agricultural production links. Its specific forms include: the application of photovoltaic in agricultural irrigation, the application of photovoltaic in farm lighting, the application of photovoltaic in agricultural products storage, the application of photovoltaic in agricultural machinery, the application of photovoltaic in agricultural pest control, etc. [6]. Agrivoltaics reflect the combination of photovoltaic power generation and agricultural production departments, the common ones are greenhouse-integrated photovoltaics, planting-integrated photovoltaics, forestry-integrated photovoltaics, animal husbandry-integrated photovoltaics, sand control-integrated photovoltaics, etc. [37,38,39]. The above two aspects compose photovoltaic agriculture. 

#### 2.2.2. The Ecosystem of Photovoltaic Agriculture

Tansley, one of the founders of the British Ecological Society, proposed the concept of “ecosystem” in 1935 [40], considering the biological and abiotic components as a whole. Similar to the various types and levels of biological systems, ecosystems are characterized by openness, and even if the overview and function of the system may remain unchanged during a certain period, the material input and output of the system remain uninterrupted. The ecosystem includes systems, an input environment, and an output environment. 

According to the classification of the ecosystem, it can be natural, such as a terrestrial ecosystem and aquatic ecosystem; It can be artificial, such as an urban ecosystem; In addition, it can also be a semi-artificial and semi-natural agricultural ecosystem [41]. Therefore, agricultural ecosystems have some characteristics of both artificial ecosystems and natural ecosystems. Such as the sociality, volatility, comprehensiveness, and selectivity of artificial ecosystems, as well as the vulnerability, nature and extreme dependence of natural ecosystems on the environment. Naturally, the input and output environment of the agricultural ecosystem must include three aspects: nature, society, and economy. The introduction of photovoltaic power generation will not change the nature of the agricultural ecosystem, and the input and output environment of the ecosystem of photovoltaic agriculture still includes the above three aspects. 

The ecosystem of photovoltaic agriculture is a multi-level, multi-functional complex ecosystem. From the perspective of a composite ecosystem, it is divided into a natural-ecological subsystem, economic-ecological subsystem, and social-ecological subsystem, as shown in Figure 3. 

##### Natural-Ecological Subsystem

The natural-ecological subsystem is a natural component of the ecosystem of photovoltaic agriculture. The ecosystem of photovoltaic agriculture can be considered as the introduction of photovoltaic power generation in the agricultural ecosystem. The generated photovoltaic power generation can be used in agricultural production, and the remaining power can be used in other aspects of human society. In other words, the introduction of photovoltaic power generation does not change the natural quality of agricultural ecosystems. At this time, photovoltaic power generation and agricultural production are conducted in the agricultural natural environment. Therefore, the natural-ecological subsystem of the ecosystem of photovoltaic agriculture is basically the same as the natural ecological ecosystem of an agricultural ecosystem. As shown in Figure 3, the natural subsystem of the ecosystem of photovoltaic agriculture includes the terrestrial ecosystem and the aquatic ecosystem. The terrestrial ecosystem includes grassland ecosystem, farmland ecosystem, forest ecosystem, and desert ecosystem, while the aquatic ecosystem includes river ecosystem, lake ecosystem, wetland ecosystem, and pond ecosystem. 

##### Economic-Ecological Subsystem

Economic-ecological subsystem is an economic organic whole composed of economic development elements, industrial sector structure, production, distribution, and consumption in the process of material circulation, energy flow, and information transmission of the ecosystem of photovoltaic agriculture. Since photovoltaic power generation and agricultural production are the two major elements of the symbiotic system of photovoltaic agriculture, the economic-ecological subsystem of the photovoltaic agricultural ecosystem naturally includes agriculture and the photovoltaic industry. Moreover, because the spatial combination of photovoltaic facilities and agriculture can produce the effect of characteristic landscape, and the characteristics of leisure and entertainment in agriculture, the economic-ecological subsystem also includes tourism. Economic-ecological subsystem provides the economic foundation for the sustainable development of the ecosystem of photovoltaic agriculture. It is the link between the natural-ecological subsystem and the social-ecological subsystem, and helps to accelerate the material and energy flow of the ecosystem of photovoltaic agriculture and the succession process of the ecosystem. 

##### Social-Ecological Subsystem

Social-ecological subsystem is a part of the ecosystem of photovoltaic agriculture that reflects social demand, input, and output. Firstly, the social-ecological subsystem should reflect the social needs of human beings, including food and clean energy. Secondly, it should reflect the relevant facilities built by human society, the technology and equipment invented and the labor force equipped, which involve both agriculture and photovoltaic. In addition, relevant policies are included to facilitate the rational allocation of resources. 

The development of the ecosystem of photovoltaic agriculture is a regulatory process with human beings as the main body and governed by the laws of natural, social, and economic development. In general, the interaction, influence, and transformation of the natural-ecological, social-ecological and economic-ecological subsystems constitute an organic whole of the ecosystem of photovoltaic agriculture.

#### 2.2.3. Concept Definition of the Ecological Niche of Photovoltaic Agriculture

In general, the ecological niche is the occupation and utilization of resources by species, and their status and function are reflected in the environment. Any organism will occupy a certain amount of resources in the environment to build its own ecological niche, so as to ensure its survival and development. The ecological niche of photovoltaic agriculture is the environmental resource occupied by the industrial organization in a certain period of time, and the function and status of the organization in the natural-economic-social environment in which it is located. Photovoltaic agriculture occupies a certain ecological niche in the dimension of environmental resources, and plays a certain function and role. Its resource living space can be divided into three main resource and environmental gradients, namely resources, technology, and policy (see Figure 4), and it has (natural) environmental, social and economic functions. 

### 2.3. Preliminary Construction of the Niche Evaluation Index System of Photovoltaic Agriculture

Photovoltaic agriculture occupies a certain amount of resources in the dimension of environmental resources, and builds its own ecological niche, so as to ensure its survival and development. Its resource living space can be divided into resources, technology, and policy. After occupying these environmental resources and building the ecological niche, they can play their functions and roles in the environmental, social, and economic aspects, thus enhancing their status. Therefore, the niche evaluation of photovoltaic agriculture can start from two aspects of environmental resources and function. The former covers resource niche, technology niche and policy niche, while the latter covers environmental niche, social niche, and economic niche. Based on the above indicators and relevant documents, the niche evaluation index system of photovoltaic agriculture is preliminarily constructed (see Table 1). 

#### 2.3.1. Resource Niche

Agricultural resources are the basic resources for the generation and development of photovoltaic agriculture, and in agricultural resources, agricultural natural resources are in the primary position. At the same time, light resources are also indispensable, because it is necessary resources for “photosynthesis” and the “photovoltaic effect”. In addition, the continuous growth of the population, the lack of resources, and environmental deterioration make the combination of photovoltaic agriculture have a broad market [42]. The emergence of photovoltaic agriculture can alleviate the above contradiction between grain production and photovoltaic power generation for land resources. At the same time, it can achieve “one land for two purposes” while protecting the environment, and meeting the needs of economic and social development. Furthermore, the construction of projects of photovoltaic agriculture requires a large amount of capital expenditures (CAPEX), and the corresponding operational expenditures (OPEX) need to be invested after the completion, especially the former accounts for a large [29]. Therefore, the development of photovoltaic agriculture needs to invest a large amount of capital resources. Finally, the operation of photovoltaic agriculture also needs personnel input, that is, human resources. 

#### 2.3.2. Technology Niche

As an intermediate organization, the role of scientific research institutions in the generation of new technologies cannot be ignored, and it plays a great role in promoting the development of new things [43]. Besides, academic conferences can help in academic exchanges, and new ideas and new technologies can be spread in this way. Furthermore, the public publication of academic papers is one of the ways of reflecting scientific research achievements, especially high-level academic papers, which are the embodiment of the most cutting-edge innovative achievements in this field. Invention patents are another way of expressing research results. Compared with academic papers, they are more closely related to productivity transformation. In addition, technological improvement can promote a better “symbiosis” of photovoltaic power generation and agricultural production, and enhance the synergistic effect between the two. Finally, the establishment of technical norms can play a quality assurance and the basis of professional management. Photovoltaic agriculture needs to establish unified technical norms to standardize the development of the industry and prevent the emergence of “non-agriculture” in the system. 

#### 2.3.3. Policy Niche

The development of photovoltaic agriculture is affected by agricultural policies, and the implementation of relevant agricultural policies is conducive to the improvement of their ecological niche. However, regarding photovoltaic agriculture, which is still in the early stage of development, its development needs more special and targeted policy support. In addition, as the projects of photovoltaic agriculture require a large amount of construction, operation, and maintenance funds, if there is an amount policy for such projects, it will contribute to the smooth development of the project. Moreover, agricultural land resources are the primary natural resources for the formation of photovoltaic agriculture. In fact, the “agriculture” in photovoltaic agriculture first refers to the nature of the land, and secondly refers to agricultural production. Therefore, the land use problem is the first problem to be solved in the development of photovoltaic agriculture, and the solution to this problem requires the corresponding land use policies. Finally, the development of photovoltaic agriculture is deeply affected by the photovoltaic industry environment and policies. 

#### 2.3.4. Environmental Niche

Saving fossil resources is a common feature of renewable energy power generation, and also an important reason for vigorously developing the renewable energy industry. Firstly, photovoltaic agriculture can generate photovoltaic electricity, reducing the consumption of fossil resources such as coal, oil, and natural gas [7]. Then, the erection of photovoltaic panels hinders the direct exposure of sunlight to the crops, thus changing the microclimate environment such as the light, temperature, and humidity of the crop growth [22]. In addition, the use of fossil resources will release small amounts of sulfur dioxide and soot, resulting in a decline in air quality. The application of photovoltaic agriculture reduces the emissions of these harmful substances and improves air quality [44]. At the same time, due to reducing the consumption of fossil resources, photovoltaic agriculture reduces carbon dioxide gas emissions and reduces the generation of the greenhouse effect. In addition, in some areas with very strong light, the construction of photovoltaic power stations reduces the excessive evaporation of water in the soil, making the originally arid land regain a certain humidity, and has the conditions for plant growth, and growing green vegetation, thus preventing the loss of water and soil [26]. 

#### 2.3.5. Social Niche

The application of photovoltaic agriculture can drive the development of smart agriculture, such as solar photovoltaic technology can be applied to farmland intelligent irrigation systems [45] and the optimal scheduling of smart agriculture greenhouse micro-energy networks [46]. Besides, energy and agriculture are the two major indispensable industries for the development of human society. Photovoltaic agriculture also meets the human demand for energy and food, and promotes the structure optimization of the energy industry and agriculture. In addition, the emergence of photovoltaic agriculture solves the demand for photovoltaic power generation for land resources, which can produce more clean energy and alleviate the demand for energy, especially clean energy [42]. Also, photovoltaic agriculture can promote food security. For example, solar photovoltaic drip irrigation technology can improve food safety in rural Sudano–Sahel areas in West Africa [5]. As well, the application of photovoltaic insecticidal lamps can solve the problem of pesticide residues in agricultural products, ensuring food safety while protecting the ecological environment. Moreover, as one of the market segments of photovoltaic power generation, photovoltaic agriculture can play a role in ensuring energy security. Finally, the development of photovoltaic agriculture can provide employment, and the construction, operation, and maintenance of various projects will all require relevant personnel, including photovoltaic practitioners and agricultural practitioners [18]. 

#### 2.3.6. Economic Niche

Many studies have proved that the application of photovoltaic agriculture can increase the land yield rate [21,47], and that the agricultural output value is an important part of the economic niche of photovoltaic agriculture. Moreover, photovoltaic agriculture can also promote the development of local related industries and improve the economic benefits of these industries. In addition to the agricultural output value and the output value of photovoltaic power generation, some projects have also added sightseeing tourism. For example, China Energy’s Dali project of photovoltaic agriculture in Shaanxi province incorporates the above elements, and tourists can visit and pick in the photovoltaic agricultural ecological park, increasing the project income. 

## 3. Materials and Methods

### 3.1. Data Sources

Through two years of research, we formed a long-term cooperative relationship with the IT Electronics Eleventh Design and Research Institute Scientific and Technological Engineering Corporation Limited, TW New Energy and GCL. This study invited 20 experts and scholars with rich experience in the field of Chinese photovoltaic agriculture to form the expert group of this research, including 5 scholars, 7 project scheme design experts, 8 project operation and maintenance leaders, and each expert has more than 5 years of working experience in photovoltaic agriculture-related fields. On the basis of the literature research, exploratory interviews were conducted with the experts and scholars. The relevant data collected in this study were distributed to the experts, so as to better clarify the scope and specific questions of the questionnaire survey, and that can also let the experts have a preliminary understanding of the investigation work of this study. In the course of the questionnaire survey (see Appendix A), the experts were introduced the purpose, the filling requirements and the contents of the questionnaire to improve the validity of the questionnaire data. The data obtained laid the data foundation for subsequent studies. 

### 3.2. Rough Set

Rough Set (RS) was first proposed by Polish scientist Pawlak in 1982 [48]. The theory is a mathematical tool that handles data by solving the upper and lower approximations. The core principle of the rough set is the attribute reduction. First of all, keep the decision attributes unchanged, and then use special methods to identify the important information, abandon the unimportant information. The details involved are as follows. 

Based on rough set theory, the knowledge system S can be characterized by a quaternion S=〈U,A,V,f〉. Where, U is a finite nonempty set, representing domain; A is an attribute set; V represents the set of attribute values; f:U×A→V is the information function formula, reflecting the relationship between elements of attribute set A and elements of attribute value V. The attribute set A of S consists of two parts, namely, conditional attribute set C and decision attribute set D, whose relationship can be expressed as A=C∪D, C∪D≠Φ. In addition, the knowledge expression system can also be expressed in the way of the decision system, that is (U,C∪D). The decision table consists of two parts: attributes and objects, which are represented by columns and rows, respectively. Therefore, a particular row in the decision table corresponds to an object information. According to the above, the decision table is a set of equivalence relations, and the attributes and the equivalence relations listed in the table correspond to each other. 

One of the key factors in rough set theory is knowledge reduction. Knowledge is important, secondary, and even irrelevant. The function of knowledge reduction is to retain the main knowledge and abandon the unimportant and irrelevant knowledge on the premise of maintaining the classification ability. The study of reduction and core is based on the following definition: order R is a clan equivalence relationship, and r is one of the equivalence relationships. If there is equation Ind(R)=Ind(R−{r}), r can be omitted in this family of equivalence relationship, and vice versa. If r can be omitted, the family of equivalence relationship R is called dependent; if r cannot be omitted, then R is independent. All attributes in the attribute set Q are indispensable. When Q⊆P and Ind(Q)=Ind(R), then Q is a reduction (Red(P)) of P. If (Red(P)) describes all the indispensable set of attributes in P, called the core of P, then the full intersection of Red(P) is equal to the core of P, record Core(P)=∩Red(P). According to the above formula, the relationship between core and reduction can be sorted out through the reduction. Core in the knowledge base has a crucial role, and this knowledge must be retained in the reduction process. 

### 3.3. DANP

In this paper, the cloud model was used to evaluate the ecological niche of photovoltaic agriculture. The niche evaluation index system of photovoltaic agriculture (see Table 1) involves many factors, and the impact of these factors on the niche level of photovoltaic agriculture is different. So, to determine the influence of each factor, this paper used the DANP method to assign different weights to each indicator. 

DANP method is an empowerment method that combines the Analytical Hierarchy Process (ANP) and DEMATEL methods. This method not only considers the network relationship between indicators, but also classifies indicators according to cause factors and result factors, and the weight of the indicators calculated is more scientific and objective. Therefore, this paper combined the calculation results of DEMEL with ANP to obtain the weights of each indicator. 

The DANP method has two steps. First, the DEMATEL method is used to build the network relationship graph, and then the DEMATEL calculation results are introduced into the DANP method to calculate the index weights. The specific steps are: 

#### 3.3.1. Determine the Network of Indicators

Step 1: Build a direct impact matrix

Determine the mutual influence relationship between various factors and construct a direct impact matrix M. The degree of direct influence between factors is expressed by 0, 1, 2, 3, 4. The specific meaning of each value is shown in Table 2. 

Construct the direct impact matrix of primary indicators and secondary indicators. If there are n factors in the network layer, then the matrix $M={\left[m_{ij}\right]}_{n\times n}$ is obtained, which $m_{ij}$ represents the degree of influence of factor i on factor j.
$$\left[\begin{array}{ccccc} m_{11} & \cdots & m_{1j} & \cdots & m_{1n}\\ \vdots & \; & \vdots & \; & \vdots \\ m_{i1} & \cdots & m_{ij} & \cdots & m_{in}\\ \vdots & \; & \vdots & \; & \vdots \\ m_{n1} & \cdots & m_{nj} & \cdots & m_{nn} \end{array}\right]$$

Step 2: Matrix normalization

The direct influence matrix is transformed to obtain a normalized matrix. Formulas (1) and (2) are used to normalize the constructed direct impact matrix M, so as to obtain the normalized matrix N.
(1)N=kM
(2)$$k=\min\left\{\frac{1}{\max_{i}\sum_{j=1}^{n}m_{ij}},\frac{1}{\max_{j}\sum_{i=1}^{n}m_{ij}}\right\}$$
where $\max_{i}\overset{n}{\underset{j=1}{\sum }}m_{ij}$ is the maximum number of rows, and $\max_{j}\overset{n}{\underset{i=1}{\sum }}m_{ij}$ is the maximum column total. 

Step 3: Calculate the comprehensive impact matrix

The normalization matrix N calculated in step 2 is used to process formula (3). The comprehensive impact matrix of the primary indicators $X_{D}$ is calculated, and the comprehensive impact matrix of the secondary indicators $X_{B}$ is calculated too.
(3)$$X=N+N^{2}+N^{3}+\cdots =N{\left(I-N\right)}^{-1}$$
where I denotes the unit matrix. 

#### 3.3.2. Determine the Weights

Step 4: Calculate the standardization matrix

Normalizing the comprehensive impact matrix $X_{D}$ calculated in step 3 according to the method shown in Equations (4) and (5), the standardization matrix $X_{D}^{\alpha }$ is calculated. 

The comprehensive impact matrix of secondary indicators $X_{C}$ is different from the standardization method of primary indicators. Each sub-matrix in the matrix is standardized one by one, and then the standardized matrix $X_{B}^{\alpha }$ is obtained. 

Taking the submatrix $X_{B}^{\alpha 23}$ as an example, the standardization process is shown in Formulas (6)–(8).
(4)$$X_{D}^{\alpha }={\left[x_{D}^{\alpha ij}\right]}_{m\times m}=\left[\begin{array}{ccccc} x_{D}^{11}/d_{1} & \cdots & x_{D}^{1j}/d_{1} & \cdots & x_{D}^{1m}/d_{1}\\ \vdots & & \vdots & & \vdots \\ x_{D}^{i1}/d_{i} & \cdots & x_{D}^{ij}/d_{i} & \cdots & x_{D}^{im}/d_{i}\\ \vdots & & \vdots & & \vdots \\ x_{D}^{m1}/d_{m} & \cdots & x_{D}^{mj}/d_{m} & \cdots & x_{D}^{mm}/d_{m} \end{array}\right]$$
(5)$$d_{i}=\sum_{j=1}^{m}t_{D}^{ij},i=1,2,\ldots ,m$$
(6)$$X_{B}^{\alpha }=\left[\begin{array}{ccccc} X_{B}^{\alpha_{11}} & \cdots & X_{B}^{\alpha_{1j}} & \cdots & X_{B}^{\alpha_{1m}}\\ \vdots & & \vdots & & \vdots \\ X_{B}^{\alpha_{i1}} & \cdots & X_{B}^{\alpha_{ij}} & \cdots & X_{B}^{\alpha_{im}}\\ \vdots & & \vdots & & \vdots \\ X_{B}^{\alpha_{m1}} & \cdots & X_{B}^{\alpha_{mj}} & \cdots & X_{B}^{\alpha_{mm}} \end{array}\right]$$
(7)$$X_{B}^{\alpha_{23}}=\begin{array}{cc} & \begin{array}{ccccc} B_{31} & \cdots & B_{3j} & \cdots & B_{3m_{3}} \end{array}\\ \begin{array}{c} B_{21}\\ \vdots \\ B_{2i}\\ \vdots \\ B_{2m_{2}} \end{array} & \left[\begin{array}{ccccc} x_{11}^{23}/x_{1}^{23} & \cdots & x_{1j}^{23}/x_{1}^{23} & \cdots & x_{1m_{3}}^{23}/x_{1}^{23}\\ \vdots & & \vdots & & \vdots \\ x_{i1}^{23}/x_{i}^{23} & \cdots & x_{ij}^{23}/x_{i}^{23} & \cdots & x_{1m_{3}}^{23}/x_{i}^{23}\\ \vdots & & \vdots & & \vdots \\ x_{m_{21}}^{23}/x_{m_{2}}^{23} & \cdots & x_{m_{2j}}^{23}/x_{m_{2}}^{23} & \cdots & x_{m_{2}m_{3}}^{23}/x_{m_{2}}^{23} \end{array}\right] \end{array}$$
(8)$$x_{i}^{23}=\sum_{j=1}^{m_{3}}x_{ij}^{23}$$

Step 5: Calculate an unweighted hypermatrix

This step transposes the normalized matrix $X_{B}^{\alpha }$ obtained in step 4 to obtain the unweighted hypermatrix $W^{*}$, as shown in Equation (9).
(9)$$W^{*}={\left(X_{B}^{\alpha }\right)}^{\prime }$$

Step 6: Calculate a weighted hypermatrix

According to the method shown in Equation (10), the weighted hypermatrix $W^{\alpha }$ is calculated.
(10)$$W^{\alpha }=X_{D}^{\alpha }W^{*}=\left[\begin{array}{ccccc} x_{D}^{\alpha_{11}}\times W_{11}^{*} & \cdots & x_{D}^{\alpha_{j1}}\times W_{i1}^{*} & \cdots & x_{D}^{\alpha_{n1}}\times W_{n1}^{*}\\ \vdots & & \vdots & & \vdots \\ x_{D}^{\alpha_{1i}}\times W_{1j}^{*} & & x_{D}^{\alpha_{ij}}\times W_{ij}^{*} & & x_{D}^{\alpha_{in}}\times W_{nj}^{*}\\ x_{D}^{\alpha_{1n}}\times W_{1n}^{*} & & x_{D}^{\alpha_{nj}}\times W_{in}^{*} & & x_{D}^{\alpha_{nn}}\times W_{nn}^{*} \end{array}\right]$$
where $W_{ij}^{*}$ is the submatrix of $W^{\alpha }$, and $x_{D}^{\alpha_{ji}}$ is the element of $X_{D}^{\alpha }$. 

Step 7: Weight calculation

The weighted hypermatrix $W^{\alpha }$ obtained in step 6 is exponentially performed, as shown in Equation (11). When the number of the multiplier approaches infinite, the resulting convergence and stable limit hypermatrix W can be obtained, and the corresponding value of each element in the matrix is the weight of the element.
(11)$$W=\lim_{h\to \infty }{\left(W^{\alpha }\right)}^{h}$$

### 3.4. Cloud Model

Cloud theory, proposed by Academician Li in 1996, can deal with the problems of ambiguity and randomness [49]. The cloud model constructed based on cloud theory has a qualitative and quantitative transformation function, and can reveal the intrinsic connection between ambiguity and randomness. The cloud model has the following advantages: the language value of the qualitative evaluation is expressed by the cloud model, which reduces the subjectivity of the evaluation; the placement of determined membership with elastic intervals is an effective tool for solving complex qualitative evaluations; the fuzzy processing of the evaluation language makes the ambiguity and the randomness can perform the quantitative operation, which can improve the accuracy of the qualitative evaluation and provide a more objective reference standard for the decision-making. Cloud model is universal in many fields. It has been widely used in natural language processing, data mining and pattern recognition research. It is also effective in the comprehensive evaluation. In the process of evaluation, cloud model can transform qualitative language value into appropriate value. Using cloud to represent evaluation set and evaluation value, it has a strong scientific and operational. 

In cloud theory, the cloud is represented by three digital features, namely expected value $E_{x}$, entropy $E_{n}$ and super entropy $H_{e}$. $C\left(E_{x},E_{n},H_{e}\right)$ is used to describe the cloud model. Here, $E_{x}$ is the expected value of the spatial distribution of cloud droplets in the theoretical domain, which can accurately transform the qualitative language into the conceptual value, which is also the central value of the theoretical domain. It is reflected in the cloud graph with the “highest point” of the cloud, and the corresponding membership degree is 1. 

Entropy $E_{n}$ in the cloud model is used to measure the uncertainty of qualitative concepts, namely the ambiguity of qualitative concepts. It can not only reflect the discrete of the droplets, but also reflect the value interval of the droplets. Mapping to the cloud map, it is the span of the cloud, which increases with the entropy value $E_{n}$. 

In short, super entropy $H_{e}$ refers to the entropy of $E_{n}$, or entropy uncertainty. As a reflection of the degree of cloud droplet dispersion, it is manifested as the thickness of the cloud drop on the cloud map. The thickness of cloud droplets gradually increases with the super entropy *H_e_* value, which also means that the randomness of membership is constantly increasing. By combining the three concepts of expectation value, entropy and super entropy organically, the cloud model effectively integrates the ambiguity and randomness of the concept, and realizes the transformation from qualitative to quantitative. 

The following are the computing steps for the cloud model: 

Step 1: calculate the expectation $E_{x}$
(12)$$E_{x}=\frac{E_{x1}+E_{x2}+\cdots +E_{xn}}{n}\;$$

Step 2: calculate the entropy $E_{n}$
(13)$$E_{n}=\sqrt{\frac{1}{n-1}\sum_{i=1}^{n}{\left(E_{xi}-E_{x}\right)}^{2}}$$

Step 3: calculate the hyperentropy $H_{e}$
(14)$$H_{e}=k$$

k is the reflection of the randomness of the evaluation. In order to control the uncertainty of the evaluation, the value is generally small, usually 0.1. At this time, it can better reflect the randomness of the evaluation and simplify the evaluation process. 

If there are n adjacent cloud models in the index system, the digital features of the integrated cloud model are as follows: (15)$$E_{x}=\frac{E_{x1}E_{n1}W_{1}+E_{x2}E_{n2}W_{2}+\cdots +E_{xn}E_{nn}W_{n}}{E_{n1}W_{1}+E_{n2}W_{2}+\cdots +E_{nn}W_{n}}$$
(16)$$E_{n}=E_{n1}W_{1}+E_{n2}W_{2}+\cdots +E_{nn}W_{n}$$
(17)$$H_{e}=k$$

## 4. Results

### 4.1. Reduction of Evaluation Indicators Based on Rough Set

The 32 niche evaluation indicators of photovoltaic agriculture selected in Table 1 were reduced to obtain a more objective and accurate index system. This paper used Rosetta software to calculate the attribute reduction of rough set. This software belongs to the KDD decision analysis system, and it is mainly based on the table logic data tool of rough set theoretical framework. 

#### 4.1.1. Establishment of Niche Evaluation Decision Table of Photovoltaic Agriculture

When evaluating the ecological niche of photovoltaic agriculture, the initial decision information table S=〈U,A,V,f〉 needs to establish firstly. Experts in the industry were invited to score the evaluation index of niche level of China’s photovoltaic agriculture as the theory domain of rough set $U=\left\{x_{1},x_{2},x_{3}\ldots \right\}$. The niche decision table consists of conditional attributes and decision attributes. The conditional attributes of this study are 32 evaluation indicators, making the conditional attribute set $C=\left\{C_{1},C_{2},C_{3},\ldots ,C_{32}\right\}=${agricultural natural resources, market resources, capital resources, light resources, human resources, scientific research institutions, academic conferences, academic papers, invention patents, technical improvement, technical norms, agriculture policy, special policies, financial policy, land-use policy, photovoltaic industry policy, conservation of fossil resources, microclimate environment improvement, air quality improvement, greenhouse gas emission reduction, conservation of water and soil, driving the development of smart agriculture, structure optimization of energy industry and agriculture, alleviating the contradiction between supply and demand for clean energy, promoting food security, ensuring energy security, providing employment, increase in land output, agricultural output value, promoting the development of related industries, output value of tourism, output value of photovoltaic power generation}, decision attribute set D={d} is the niche level. 

Experts were invited to participate in the questionnaire filling (see Questionnaire 1 of Appendix A), sort out the questionnaire data, and quantify the categories of the niche evaluation indicators of photovoltaic agriculture in the evaluation decision information table, which is the decision attribute of the rough set. 

#### 4.1.2. Attribute Reduction

The decision information of niche evaluation indicators of photovoltaic agriculture was analyzed, and the values of resource niche, technology niche, policy niche, environmental niche, social niche, and economic niche are successively input into the processing program, so that the attributes of niche evaluation indicators of photovoltaic agriculture can be simplified. The Genetic algorithm and Johnson’s algorithm in Rosetta software were used, and the specific reduction results are shown in Table 3. 

#### 4.1.3. Construction of Final Niche Evaluation Index System of Photovoltaic Agriculture

Screening the indicators using rough set, redundancy indicators are light resources (C_4_) of resource niche, academic conferences (C_7_) and technical improvement (C_10_) of technology niche, financial policy (C_14_) of policy niche, air quality improvement (C_19_) of environmental niche, promoting food security(C_25_) and providing employment (C_27_) of social niche, output value of tourism (C_31_) of economic niche. The accuracy of the decision rules obtained after attribute reduction is all 100%. The optimized niche evaluation index system of photovoltaic agriculture after deleting the redundancy indicators is shown in Table 4. 

As can be seen from Table 4, the final niche evaluation index system of photovoltaic agriculture is divided into two categories: environmental resources and functional status. It is then divided into six primary indicators, named resource niche, technology niche, policy niche, environmental niche, social niche and economic niche. There are four secondary indicators under each primary index, specifically, resource niche includes including agricultural natural resources, market resources, capital resources, human resources. Technology niche includes scientific research institutions, academic papers, invention patents, technical norms. Policy niche including agricultural policy, special policy, land use policy, photovoltaic industry policy. Environmental niche includes fossil resource conservation, microclimate environment improvement, greenhouse gas emissions reduction, conservation of water and soil. Social niche includes driving the development of smart agriculture, structure optimization of energy industry and agriculture, alleviating the contradiction between energy supply and demand for clean energy, ensuring energy security. Economic niche includes increase in land output, agricultural output value, promoting the development of related industries, output value of photovoltaic power generation. 

### 4.2. Weight Determination Based on DANP

First, the comprehensive influence matrix was obtained using the DEMATEL method. This paper invited experts to distinguish the strength of the correlation between the various elements (line elements for column elements) in the niche evaluation index system (see Questionnaire 2 of Appendix A). A value of 0 is no effect, 1 is very weak, 2 is weak, 3 is strong, 4 is very strong. The direct influence matrix was obtained by averaging the expert scoring data. The normalized matrix can be obtained by using Formulas (1) and (2) to calculate the direct influence matrix. The index system is composed of six primary indicators and 24 secondary indicators, which obtaining the normalized matrix of primary and secondary indicators respectively. According to the resulting normalized matrix, the comprehensive influence matrix can be further calculated according to Formula (3). 

Secondly, the weight of each indicator was calculated. In this section, the weights of indicators were calculated using the DANP method based on the comprehensive influence matrix obtained by the DEMATEL method. Standardized matrix $X_{D}^{\alpha }$ of primary indicators was calculated according to Formulas (4) and (5), and standardized matrix $X_{B}^{\alpha }$ of secondary indicators was calculated according to Formulas (6)–(8). According to Formulas (9) and (10), the unweighted hypermatrix $W^{*}$ and the weighted hypermatrix $W^{\alpha }$ of the indicators of niche evaluation index system of photovoltaic agriculture were obtained, the latter is shown in Table 5. 

Finally, according to Formula (11), the limit $\underset{h\to \infty }{lim}{\left(w^{\alpha }\right)}^{h}$ of the weighted supermatrix $W^{\alpha }$ was calculated, and the limit hypermatrix W was obtained. The weight of each indicator is shown in Table 6. Wherein, weight 1 refers to the weight of the secondary indicator for the integrated niche, and weight 2 refers to the weight of the secondary indicator against the corresponding primary indicator. 

### 4.3. Niche Evaluation Based on the Cloud Model

#### 4.3.1. Indicator Comment Set

This paper established a comment set on the evaluation indices: V = very low, low, medium, high, very high. We invited experts to score the comment set between 0 and 1 (see Questionnaire 3 of Appendix A). Then the expectation, entropy, and super entropy of every class of niche level were calculated. We used Matlab to convert the eigenvalues of the comment set to a cloud map, as shown in Figure 5. 

#### 4.3.2. Niche Evaluation of Photovoltaic Agriculture

The data of questionnaire (see Questionnaire 4 of Appendix A) and the weights calculated in Section 4.2 were inserted into Formulas (12)–(14) to obtain the expected value, entropy and super-entropy of each primary indicator, and then the digital eigenvalues of indicators were calculated according to Formulas (15)–(17), see Table 7.

Using Matlab software, the eigenvalues of each indicator in Table 7 are represented in the form of cloud chart, as shown in Figure 6 and Figure 7.

As can be seen from the cloud map, the resource niche is between medium and high levels, while the rest of the niches are all between low and medium levels. Among them, the technology niche, social niche and economic niche are at relatively low levels, while the policy niche and environmental niche are at a relatively medium level. On the whole, the ecological niche of China’s photovoltaic agriculture is between a low to medium level.

## 5. Discussion and Conclusions

### 5.1. Discussion

#### 5.1.1. The Level of Resource Niche

As shown in Figure 5, in all aspects of the comprehensive niche, the resource niche has the highest expected values, is between the medium and high levels, and is slightly biased towards the medium level. From the perspective of the specific factors affecting the resource niche, the agricultural natural resources are at a relatively high level, the market resources are close to a relatively high level, and both the capital resources and human resources are at a lower medium level. The reason is that China is a big agricultural country, with rich agricultural natural resources and economic resources. Nowadays, China is vigorously developing the new energy industry, and agriculture is an indispensable industry for human survival. Coupled with the limitation of land resources, photovoltaic agriculture has a good market foundation. In addition, due to the large investment in the construction and operation and maintenance of photovoltaic facilities, the capital resources are relatively scarce. In terms of human resources, there is a lack of compound talents who understand both photovoltaic technology and agricultural production.

#### 5.1.2. The Level of Technology Niche

The technology niche is between the low and medium levels and slightly biased towards the low level. As for the influencing factors of the technology niche, the four aspects are located in the low to medium level range. Compared with the French National Academy of Agricultural Sciences and the Fraunhofer Institute of Solar Energy Systems in Germany, China still lacks such specialized research institutions to conduct long-term, systematic and comprehensive research on photovoltaic agriculture. In terms of academic papers, compared with the leading projects of photovoltaic agriculture in China, the number of influential papers is relatively insufficient. Invention patents and technical norms are slightly stronger, and some enterprises are promoting the application for invention patents, such as the GCL New Energy mentioned above. In terms of technical norms, China has made some progress, but mainly for a few areas on the individual mode of photovoltaic agriculture.

#### 5.1.3. The Level of Policy Niche

The policy niche is near the medium level. Among the four aspects that affect the policy niche, the expected values of the photovoltaic industry policy are relatively high. It is the policy support that has promoted the progress of photovoltaic technology and the development of the photovoltaic industry, which are the main driving forces for the formation and development of photovoltaic agriculture. However, the direct promotion of photovoltaic agriculture development must be special policy support, and China currently has almost no such policy. The land use policy is more relaxed than before. The agriculture and forestry departments in some regions have approved a considerable number of project land in the past two years, as illustrated by the situation in Guizhou province in 2020 and 2021. Recent agricultural policies are beneficial to the development of photovoltaic agriculture, such as high-quality agricultural development policy and photovoltaic poverty alleviation policy.

#### 5.1.4. The Level of Environmental Niche

The environmental niche is between the low and medium levels and is biased towards the medium level. In terms of fossil resource conservation and greenhouse gas emission reduction, photovoltaic agriculture is not much different from other types of photovoltaic systems. However, the unique function of the system mainly lies in soil and water conservation and improving the microclimate environment of crop growth. The former is mainly reflected in the projects in arid or semi-arid areas, such as the sand control-integrated photovoltaic projects mentioned above, in which China has achieved some results. The latter is mainly reflected in the greenhouse-integrated and planting-integrated photovoltaic projects, but due to the limited technology, the role of this aspect still needs to be improved.

#### 5.1.5. The Level of Social Niche

The social niche is also between the low and medium levels, and its level is relatively close to the level of the economic niche. Since the application of photovoltaic in agriculture can solve the problem of power consumption in agriculture [50], especially in remote areas, and the power output from the agrivoltaic system can be applied to all aspects of human society, photovoltaic agriculture plays a more prominent role in alleviating the contradiction between supply and demand of clean energy than the other three aspects of the social niche. Accordingly, photovoltaic agriculture can ensure energy security to a certain extent. In terms of structure optimization of energy industry and agriculture, photovoltaic agriculture can play a direct role. The premise is to give full play to the synergistic effect of photovoltaic power generation and agricultural production, and there is still a lot of room for improvement in this respect. In addition, the development of smart agriculture should also be based on the development of the above synergies.

#### 5.1.6. The Level of Economic Niche

According to the calculation results in Table 7, photovoltaic agriculture has not yet played a good economic function. As mentioned above, in practice, photovoltaic agriculture often tends to favor photovoltaic power generation and ignores agricultural production. In fact, agricultural output value is crucial to the realization of the economic effect of photovoltaic agriculture. Only when the economic value of agriculture is fully reflected, can the economy of photovoltaic agriculture possibly be better than other photovoltaic systems. Of course, this also depends on the synergies of photovoltaic power generation and agricultural production. In some cases, if the project (generally the agricultural part) can be combined with the local related industries, it can drive the development of related industries [51]. But in the industry environment of “emphasizing photovoltaic over agriculture”, this role is difficult to be reflected. In addition, the increase in land output will also need to be based on both photovoltaic and agricultural production.

#### 5.1.7. The Level of Integrated Niche

As shown in Figure 6, the comprehensive niche of photovoltaic agriculture in China is between the low and medium levels, slightly biased to the medium level. It shows that China’s photovoltaic agriculture has a certain niche after more than ten years of development, but it is still at a lower medium level. The comprehensive niche has six aspects. Among them, only the resource niche level is between the medium to high levels, and the rest is between the low to medium levels, making the integrated niche ultimately in this level range. It should be noted that although photovoltaic agriculture occupies agricultural land, it will not hinder agricultural production. On the contrary, as long as the relationship between photovoltaic power generation and agricultural production is properly handled, photovoltaic power generation can promote agricultural production. At the same time, the clean electricity generated by photovoltaic agriculture can be consumed by the society, resulting in corresponding economic, social and environmental benefits. Therefore, this paper believes that photovoltaic agriculture makes full use of agricultural land resources, and its high niche level will be better. Of course, a high niche level of photovoltaic agriculture does not mean occupying as much agricultural land as possible, but giving full play to the synergistic effect of photovoltaic power generation and agricultural production. It shows that the development of photovoltaic agriculture in China has a good resource base. If the corresponding technological progress and policy formulation can play better environmental, social and economic functions on this basis, a higher level of ecological niche can be reached.

### 5.2. Conclusions

Firstly, based on the concept of the ecological niche of photovoltaic agriculture, this paper preliminarily selected 32 indicators from the two aspects of environmental resource and functional status to construct the niche evaluation index system of photovoltaic agriculture. Then, the rough set theory was used to simplify the preliminarily constructed index system, and 24 indicators were selected to form the final index system. Finally, the ecological niche of photovoltaic agriculture was evaluated based on the DANP method and cloud model. Overall, the niche level of China’s photovoltaic agriculture is between the low and medium levels. From the six aspects that constitute the integrated niche, the level of resource niche is the highest, between the medium and high levels, followed by the policy niche, near the medium level, then the environmental niche, which is at a slightly lower medium level; the last three in turn are technology niche, social niche, and economic niche. Among them, the levels of the last two are related with the level of technology niche. If the technology can fully realize the synergistic effect of photovoltaic power generation and agricultural production in photovoltaic agriculture, and then good policy formulation can play better environmental, social and economic functions on this basis, and a higher level of niche can be achieved.

## Figures and Tables

**Figure 1 ijerph-19-14702-f001:**
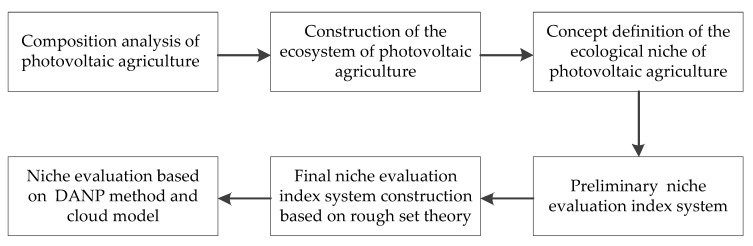
Analytical framework.

**Figure 2 ijerph-19-14702-f002:**
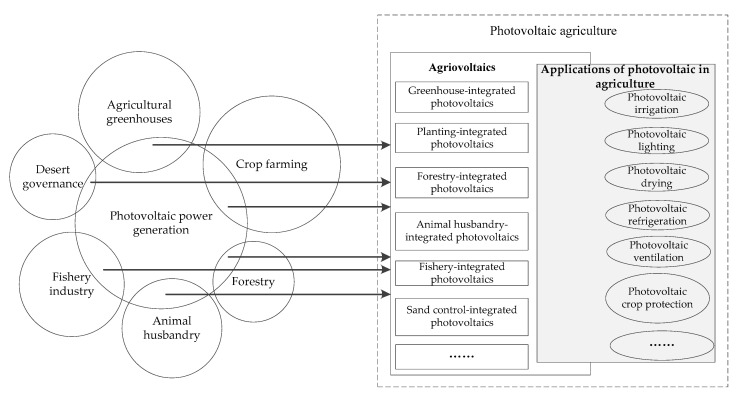
Composition of photovoltaic agriculture.

**Figure 3 ijerph-19-14702-f003:**
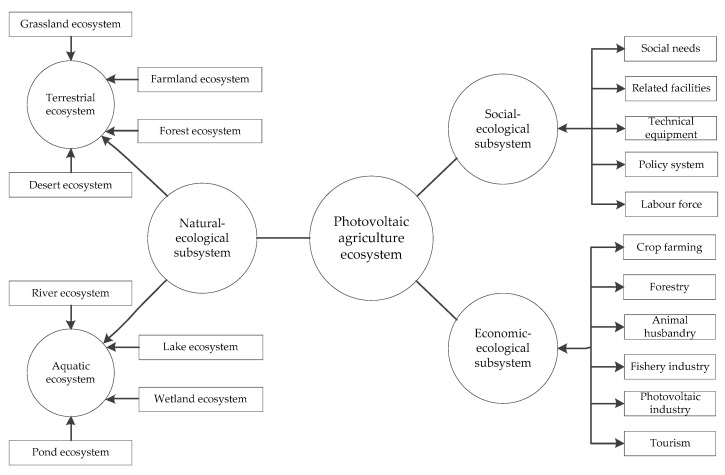
Structure of the ecosystem of photovoltaic agriculture.

**Figure 4 ijerph-19-14702-f004:**
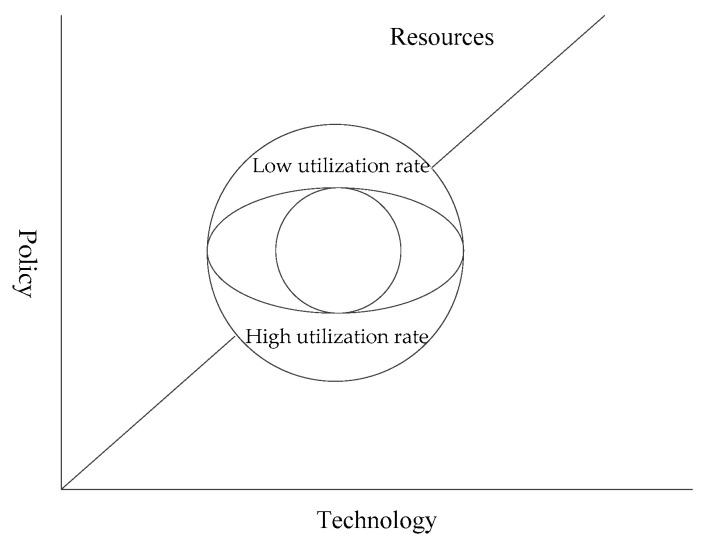
Hypervolume model of the ecological niche of photovoltaic agriculture.

**Figure 5 ijerph-19-14702-f005:**
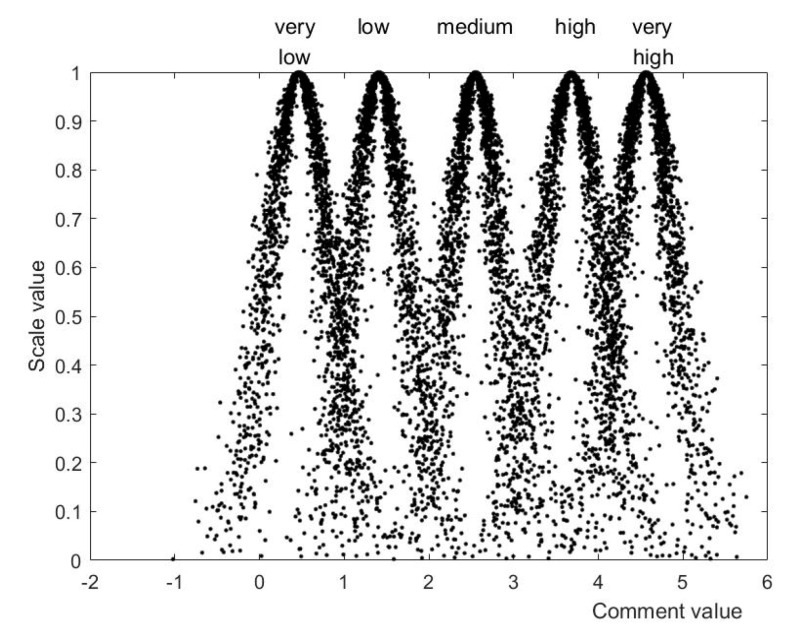
Cloud map of comment set.

**Figure 6 ijerph-19-14702-f006:**
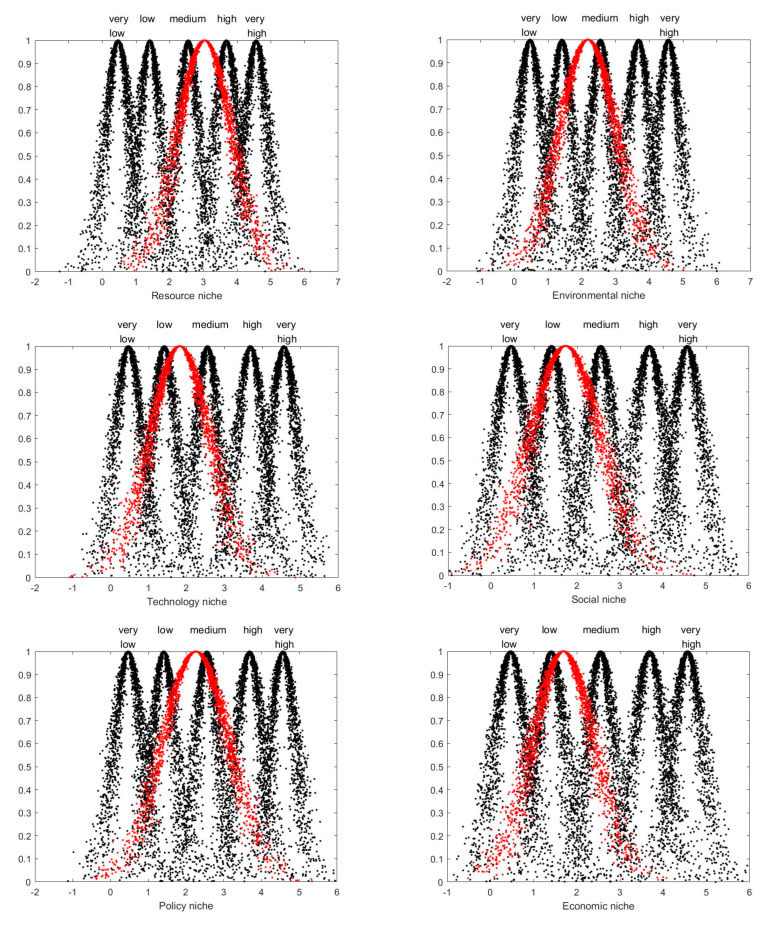
Niche cloud map of secondary each indicator.

**Figure 7 ijerph-19-14702-f007:**
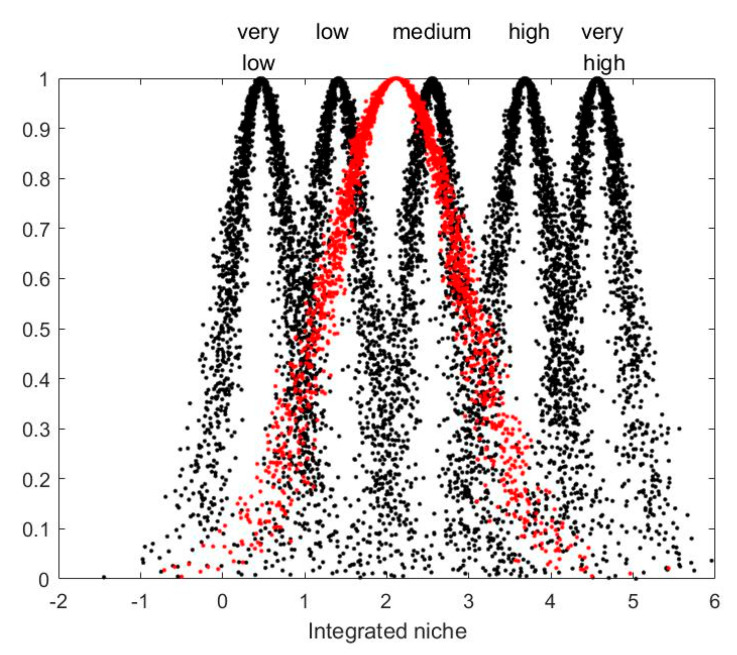
Cloud map of integrated niche.

**Table 1 ijerph-19-14702-t001:** Preliminary niche evaluation index system of photovoltaic agriculture.

Primary Indicators	Secondary Indicators	Primary Indicators	Secondary Indicators
Resource niche	Agricultural natural resources	Environmental niche	Conservation of fossil resources
	Market resources		Microclimate environment improvement
	Capital resources		Air quality improvement
	Light resources		Greenhouse gas emission reduction
	Human resources		Conservation of water and soil
Technology niche	Scientific research institutions	Social niche	Driving the development of smart agriculture
	Academic conferences		Structure optimization of the energy industry and agriculture
	Academic papers		Alleviating the contradiction between supply and demand for clean energy
	Invention patents		Promoting food security
	Technical improvement		Ensuring energy security
	Technical norms		Providing employment
Policy niche	Agriculture policy	Economic niche	Increase in land output
	Special policy		Agricultural output value
	Financial policy		Promoting the development of related industries
	Land use policy		Output value of tourism
	Photovoltaic industry policy		Output value of photovoltaic power generation

**Table 2 ijerph-19-14702-t002:** Score table of influence relationship.

Score	Meaning (Row Element to Column Element)
0	No effect
1	Very weak effect
2	Weak effect
3	Strong effect
4	Very strong effect

**Table 3 ijerph-19-14702-t003:** Results of attribute reduction.

Primary Indicators	Secondary Indicators	Genetic Algorithm	Johnson’s Algorithm	Redundancy Indicators	Reduction	Support (%)	Length
Resource niche	{C_1_,C_2_,C_3_,C_4_,C_5_}	{C_1_,C_2_,C_3_,C_5_}	{C_1_,C_2_,C_3_,C_5_}	C_4_	{C_1_,C_2_,C_3_,C_5_}	100	4
		{C_2_,C_3_,C_4_,C_5_}					
Technology niche	{C_6_,C_7_,C_8_,C_9_,C_10_,C_11_}	{C_6_,C_8_,C_10_,C_11_}	{C_6_,C_8_,C_9_,C_11_}	C_7_,C_10_	{C_6_,C_8_,C_9_,C_11_}	100	4
		{C_8_,C_9_,C_10_,C_11_}					
		{C_6_,C_8_,C_9_,C_11_}					
Policy niche	{C_12_,C_13_, C_14_,C_15_,C_16_}	{C_12_,C_13_,C_15_,C_16_}	{C_12_,C_13_,C_15_,C_16_}	C_14_	{C_12_,C_13_,C_15_,C_16_}	100	4
Environmental niche	{C_17_,C_18_,C_19_,C_20_,C_21_}	{C_17_,C_18_,C_20_,C_21_}	{C_17_,C_18_,C_20_,C_21_}	C_19_	{C_17_,C_18_,C_20_,C_21_}	100	4
		{C_18_,C_19_,C_20_,C_21_}					
Social niche	{C_22_,C_23_,C_24_,C_25_,C_26_,C_27_}	{C_22_,C_23_,C_24_,C_26_}	{C_22_,C_23_,C_24_,C_26_}	C_25_,C_27_	{C_22_,C_23_,C_24_,C_26_}	100	4
		{C_23_,C_24_,C_26_,C_27_}					
Economic niche	{C_28_,C_29_,C_30_,C_31_,C_32_}	{C_28_,C_29_,C_31_,C_32_}	{C_28_,C_29_,C_30_,C_32_}	C_31_	{C_28_,C_29_,C_30_,C_32_}	100	4
		{C_28_,C_29_,C_30_,C_32_}					

**Table 4 ijerph-19-14702-t004:** Final niche evaluation index system of photovoltaic agriculture.

Primary Indicators	Secondary Indicators	Primary Indicators	Secondary Indicators
Resource niche	Agricultural natural resources	Environmental niche	Conservation of fossil resources
	Market resources		Microclimate environment improvement
	Capital resources		Greenhouse gas emission reduction
	Human resources		Conservation of water and soil
Technology niche	Scientific research institutions	Social niche	Driving the development of smart agriculture
	Academic papers		Structure optimization of energy industry and agriculture
	Invention patents		Alleviating the contradiction between supply and demand for clean energy
	Technical norms		Ensuring energy security
Policy niche	Agriculture policy	Economic niche	Increase in land output
	Special policy		Agricultural output value
	Land use policy		Promoting the development of related industries
	Photovoltaic industry policy		Output value of photovoltaic power generation

**Table 5 ijerph-19-14702-t005:** Weighted hypermatrix of secondary indicators.

	B_11_	B_12_	B_13_	B_14_	B_21_	B_22_	B_23_	B_24_	B_31_	B_32_	B_33_	B_34_	B_41_	B_42_	B_43_	B_44_	B_51_	B_52_	B_53_	B_54_	B_61_	B_62_	B_63_	B_64_
B_11_	0.030	0.035	0.032	0.036	0.037	0.043	0.043	0.043	0.039	0.042	0.042	0.037	0.044	0.044	0.042	0.044	0.045	0.046	0.045	0.044	0.044	0.040	0.043	0.045
B_12_	0.035	0.028	0.037	0.036	0.044	0.042	0.042	0.042	0.038	0.039	0.038	0.040	0.042	0.039	0.040	0.039	0.044	0.041	0.044	0.043	0.039	0.039	0.042	0.040
B_13_	0.044	0.041	0.032	0.044	0.049	0.048	0.048	0.047	0.047	0.044	0.044	0.045	0.049	0.052	0.051	0.052	0.050	0.051	0.050	0.049	0.049	0.048	0.044	0.046
B_14_	0.035	0.040	0.043	0.028	0.048	0.045	0.045	0.045	0.045	0.045	0.045	0.047	0.046	0.046	0.048	0.046	0.048	0.048	0.048	0.050	0.038	0.043	0.042	0.040
B_21_	0.040	0.037	0.036	0.039	0.024	0.031	0.033	0.031	0.043	0.043	0.040	0.042	0.036	0.031	0.033	0.037	0.036	0.038	0.039	0.036	0.034	0.035	0.037	0.038
B_22_	0.046	0.045	0.046	0.044	0.038	0.033	0.037	0.038	0.046	0.049	0.048	0.048	0.044	0.042	0.043	0.040	0.044	0.043	0.045	0.044	0.040	0.043	0.042	0.041
B_23_	0.041	0.042	0.044	0.042	0.036	0.034	0.027	0.034	0.046	0.044	0.046	0.045	0.038	0.039	0.040	0.040	0.041	0.038	0.040	0.041	0.040	0.040	0.039	0.038
B_24_	0.040	0.042	0.041	0.042	0.034	0.034	0.035	0.028	0.043	0.043	0.046	0.045	0.036	0.042	0.038	0.037	0.041	0.043	0.039	0.041	0.043	0.040	0.039	0.041
B_31_	0.041	0.043	0.043	0.041	0.041	0.041	0.041	0.041	0.037	0.037	0.037	0.029	0.041	0.045	0.043	0.044	0.041	0.042	0.041	0.045	0.041	0.040	0.043	0.043
B_32_	0.044	0.043	0.043	0.045	0.040	0.040	0.041	0.041	0.036	0.028	0.036	0.035	0.044	0.044	0.043	0.044	0.043	0.039	0.041	0.042	0.041	0.042	0.040	0.043
B_33_	0.042	0.042	0.042	0.040	0.039	0.039	0.040	0.040	0.035	0.034	0.026	0.034	0.042	0.040	0.045	0.042	0.039	0.040	0.042	0.040	0.039	0.041	0.039	0.036
B_34_	0.043	0.042	0.042	0.044	0.040	0.040	0.038	0.038	0.026	0.036	0.036	0.037	0.046	0.044	0.042	0.043	0.042	0.044	0.040	0.038	0.040	0.038	0.039	0.039
B_41_	0.048	0.048	0.046	0.044	0.046	0.039	0.045	0.044	0.045	0.047	0.045	0.043	0.029	0.041	0.036	0.037	0.038	0.040	0.037	0.040	0.044	0.038	0.046	0.049
B_42_	0.049	0.050	0.051	0.051	0.055	0.059	0.055	0.054	0.051	0.053	0.050	0.049	0.044	0.033	0.045	0.041	0.043	0.045	0.046	0.045	0.052	0.056	0.048	0.050
B_43_	0.054	0.054	0.052	0.058	0.052	0.056	0.053	0.052	0.053	0.047	0.052	0.051	0.045	0.044	0.036	0.046	0.055	0.053	0.054	0.053	0.050	0.053	0.049	0.051
B_44_	0.052	0.052	0.054	0.051	0.051	0.049	0.051	0.053	0.047	0.049	0.050	0.053	0.040	0.040	0.041	0.034	0.053	0.052	0.053	0.051	0.048	0.047	0.051	0.045
B_51_	0.037	0.038	0.038	0.038	0.036	0.036	0.038	0.035	0.032	0.030	0.032	0.030	0.042	0.040	0.043	0.040	0.024	0.036	0.035	0.033	0.034	0.036	0.038	0.045
B_52_	0.042	0.039	0.042	0.041	0.043	0.040	0.042	0.041	0.035	0.037	0.036	0.033	0.046	0.047	0.043	0.044	0.035	0.028	0.035	0.039	0.047	0.051	0.051	0.046
B_53_	0.036	0.038	0.037	0.040	0.039	0.038	0.037	0.040	0.040	0.041	0.041	0.042	0.042	0.039	0.043	0.039	0.037	0.032	0.026	0.032	0.049	0.044	0.041	0.044
B_54_	0.043	0.044	0.041	0.039	0.042	0.046	0.044	0.044	0.043	0.042	0.041	0.045	0.042	0.046	0.042	0.049	0.037	0.038	0.037	0.030	0.054	0.053	0.054	0.049
B_61_	0.040	0.040	0.034	0.039	0.045	0.045	0.043	0.043	0.041	0.041	0.041	0.041	0.040	0.044	0.042	0.035	0.040	0.041	0.040	0.037	0.026	0.036	0.036	0.035
B_62_	0.040	0.041	0.044	0.043	0.046	0.045	0.044	0.046	0.051	0.051	0.045	0.051	0.047	0.045	0.046	0.048	0.046	0.042	0.047	0.043	0.037	0.030	0.036	0.041
B_63_	0.039	0.039	0.039	0.038	0.041	0.041	0.039	0.040	0.040	0.043	0.043	0.044	0.037	0.037	0.038	0.040	0.039	0.040	0.040	0.042	0.034	0.035	0.025	0.032
B_64_	0.038	0.038	0.041	0.037	0.036	0.036	0.041	0.038	0.039	0.036	0.042	0.035	0.038	0.036	0.037	0.039	0.038	0.039	0.036	0.041	0.036	0.031	0.034	0.025

**Table 6 ijerph-19-14702-t006:** Weights of secondary indicators.

Primary Indicators	Secondary Indicators	Weight 1	Weight 2	Primary Indicators	Secondary Indicators	Weight 1	Weight 2
Resource niche	B_11_	0.0440	0.2566	Environmental niche	B_41_	0.0362	0.2263
	B_12_	0.0410	0.2391		B_42_	0.0411	0.2569
	B_13_	0.0397	0.2315		B_43_	0.0391	0.2444
	B_14_	0.0468	0.2729		B_44_	0.0436	0.2725
Technology niche	B_21_	0.0422	0.2218	Social niche	B_51_	0.0394	0.2263
	B_22_	0.0488	0.2564		B_52_	0.0440	0.2569
	B_23_	0.0508	0.2669		B_53_	0.0387	0.2444
	B_24_	0.0485	0.2549		B_54_	0.0371	0.2725
Policy niche	B_31_	0.0362	0.2280	Economic niche	B_61_	0.0410	0.2540
	B_32_	0.0429	0.2702		B_62_	0.0410	0.2540
	B_33_	0.0399	0.2513		B_63_	0.0393	0.2435
	B_34_	0.0398	0.2506		B_64_	0.0401	0.2485

**Table 7 ijerph-19-14702-t007:** Digital eigenvalues of each indicator.

Primary Indicators	Secondary Indicators	Expected Value	Entropy	Expected Value	Entropy	Super Entropy
Resource niche	B_11_	3.75	0.8507	3.0360	0.7916	0.1
	B_12_	3.5	0.8272			
	B_13_	2.5	0.7609			
	B_14_	2.3	0.7327			
Technology niche	B_21_	1.6	0.821	1.8182	0.7844	0.1
	B_22_	1.85	0.745			
	B_23_	1.9	0.788			
	B_24_	1.9	0.788			
Policy niche	B_31_	2.9	0.8522	2.2597	0.8867	0.1
	B_32_	0.8	0.7678			
	B_33_	2.1	0.9119			
	B_34_	3.1	1.0208			
Environmental niche	B_41_	2.1	0.8522	2.1812	0.8433	0.1
	B_42_	1.5	0.7609			
	B_43_	2.1	0.9119			
	B_44_	2.9	0.8522			
Social niche	B_51_	1.4	0.8208	1.7334	0.8255	0.1
	B_52_	1.5	0.7609			
	B_53_	2.1	0.9119			
	B_54_	1.85	0.8127			
Economic niche	B_61_	1.9	0.6407	1.6872	0.7111	0.1
	B_62_	1.5	0.6882			
	B_63_	1.3	0.8013			
	B_64_	2.1	0.7182			
Integrated niche				2.1141	0.8085	0.1

## Data Availability

In the results section, data supporting reported results can be found.

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
