# Peer review of "Research on Niche Evaluation of Photovoltaic Agriculture in China"

_ijerph, 2022, doi:10.3390/ijerph192214702_

Round 1
Reviewer 1 Report
The article " Research on Niche Evaluation of Photovoltaic Agriculture in China" is interesting because the combination of photovoltaics and agriculture is new trend. The research method is appropriate however authors should provide and clarify some things as below:
1. Photovoltaic Agriculture have many advantages in environment aspects however there is still drawback that is land appropriation. So that medium level of China's photovoltaic agriculture maybe better than high lever. The authors should be clarify and discuss about this matter, which level is better.
2. the detail on the 32 indicators (data) should be provide in a supplementary which attached with this manuscript. Survey questions in section 3.1 should be provide in supplementary also.
3. It is better if the authors provide some rough research in other country for comparison (optional).
Author Response
Point 1:Photovoltaic Agriculture have many advantages in environment aspects however there is still drawback that is land appropriation. So that medium level of China's photovoltaic agriculture maybe better than high lever. The authors should be clarify and discuss about this matter, which level is better.
Response 1: It is really a very good suggestion. According to the suggestion, we clarify and discuss about this matter in the revision (see Attachment, lines 689-699, highlighted in yellow).
Point 2: The detail on the 32 indicators (data) should be provide in a supplementary which attached with this manuscript. Survey questions in section 3.1 should be provide in supplementary also.
Response 2: According to the problem pointed out, we provide the detail on the 32 indicators (data) and survey questions in section 3.1 by means of the attachment of revised manuscript.
Point 3:It is better if the authors provide some rough research in other country for comparison (optional).
Response 3: Thanks for the suggestion. We will strengthen this in future research.

Reviewer 2 Report
This paper makes some interesting points. There is immense interest in using agricultural spaces to produce solar power. The article provides some new insights on that process – including how the power generated could be used in the agricultural operations.
My critiques are more related to how the material is presented, rather than the material itself. I think the findings are fine as far as they go – especially since this appears to be an introduction to the subject, And it uses different theoretical foundations than what would be expected – niche evaluation, cloud modeling -- to examine the subject.
I feel it adds to the discussion on the subject and once the presentation is put into a better form that it is ready for publication consideration.

Author Response
Point 1: This sentence is constructed a bit awkwardly. It appears the authors predicted that photovoltaic applications would be more readily used/available/possible after 2000 -- as they were writing in 1978. (The way the sentence is structured, the year reference reads like a time of day reference.)
Response 1: According to the problem pointed out, we rewrite this sentence in the revision (see Attachment, lines 31 to 32 highlighted in yellow).
Point 2: This reads like you are asserting that growing crops among the solar panels is unique to China -- which you note above is happening in other places. So while there may be something unique in this setting to China, it is not clear what it is from the context what that is. It may be that you define exactly what this entails in Section 2. If that is so, it needs to be noted as such when you introduce the concept.
Response 2: According to the problem pointed out, we note that the concept of photovoltaic agriculture is defined exactly in Section 2. (see Attachment, lines 65 to 66, highlighted in yellow).
Point 3: You should try to avoid starting of each paragraph with the same words ("First of all") as it could confuse the reader. Especially when the phrase "First of all" is unnecessary. Related to that, "Secondly" is also overused. Some variation in the word choice will increase the readability of the paper.
Response 3: According to the problems pointed out, we improve some expressions in the revision (see Attachment, lines 230, 233, 239, 246, 248, 261, 265, 267, 279, 292, 295, 311, 313, highlighted in yellow).
Point 4: Bad page break.
Response 4: According to the problem pointed out, we move Table 2 to Page 9 in the revision (see Attachment, line 319, highlighted in yellow).
Point 5: Awkward construction.
Response 5: According to the problem pointed out, we rewrite the sentence in the revision (see Attachment, lines 497-499, highlighted in yellow).
Point 6: Bad page break.
Response 6: According to the problem pointed out, we move Table 4 to Page 14 in the revision (see Attachment, line 540, highlighted in yellow).
Point 7: Bad page break.
Response 7: According to the problem pointed out, we move Table 5 to Page 15 in the revision (see Attachment, line 582, highlighted in yellow).
Point 8: Bad page break.
Response 8: According to the problem pointed out, we move Figure 6 to Page 17 in the revision (see Attachment, line 609, highlighted in yellow).
